# Transcriptome Sequencing and Differential Analysis of Testes of 1-Year-Old and 3-Year-Old Kazakh Horses

**DOI:** 10.3390/biology15010100

**Published:** 2026-01-04

**Authors:** Jiahao Liu, Yuting Yang, Liuxiang Wen, Mingyue Wen, Yaqi Zeng, Wanlu Ren, Xinkui Yao

**Affiliations:** 1College of Animal Science, Xinjiang Agricultural University, Urumqi 830052, China; ljh072412@163.com (J.L.); 18188549491@163.com (Y.Y.); 13890451520@163.com (L.W.); 18280584310@163.com (M.W.); zengyaqi@xjau.edu.cn (Y.Z.); 2Xinjiang Key Laboratory of Equine Breeding and Exercise Physiology, Urumqi 830052, China

**Keywords:** Kazakh horse, testes, mRNA, DEGs

## Abstract

This study investigated testicular developmental changes in Kazakh horses during sexual maturation. Transcriptome sequencing comparing juvenile (1-year-old) and sexually mature (3-year-old) Kazakh horses revealed numerous differentially expressed genes (DEGs). These genes play crucial roles in spermatogenesis, testicular development, and hormonal regulation. Additionally, we identified key signaling pathways associated with testicular development, including Focal adhesion, Pathways in cancer, and the PI3K-Akt signaling pathway. These findings elucidate the biological changes in Kazakh horses’ testicular development and provide a new research foundation for genetic breeding and reproductive performance enhancement in equids.

## 1. Introduction

As an important indigenous horse breed in China, the Kazakh horse is characterized by stable genetic traits, tolerance to coarse feed, and strong stress resistance [1,2]. Due to their high genetic diversity and stable breed characteristics, Kazakh horses are widely utilized in breeding improvement, recreational riding, and livestock product production. They constitute an indispensable component of equine germplasm resources and serve as a vital genetic resource for horse breeding and selection [3,4]. In recent years, the Kazakh horse population has been declining annually due to increased horse meat consumption. Therefore, enhancing the reproductive efficiency of Kazakh horses can promote the development and conservation of the breed.

The testes are a highly complex organ in male mammals, exhibiting significant cellular heterogeneity and possessing the capacity to produce sperm and secrete androgens [5]. In normal male animals, the testes exist in pairs, presenting a twisted ellipsoid shape and are located within the two chambers of the scrotum [6,7]. They are primarily composed of seminiferous tubules and testicular interstitium [8]. Prior to the onset of sexual differentiation, testicular growth proceeds slowly, primarily involving seminiferous tubule development [9]. During puberty, testicular growth accelerates as seminiferous tubules gradually differentiate into convoluted seminiferous tubules, initiating spermatogenesis [10]. Upon reaching puberty, the interstitial tissue rapidly increases, spermatogenesis within the seminiferous tubules accelerates, and testicular volume expands significantly [11]. During sexual maturity, testicular volume continues to grow, mature sperm appear abundantly in the tubular lumen, and spermatogonia enlarge and gradually migrate toward the luminal side [12]. The reproductive capacity of male animals is influenced by multiple factors, including mating ability, libido intensity, timing of sexual maturation, and semen quality. These factors are closely related to the developmental status of the testes [13]. Therefore, normal testicular development plays a crucial role in the reproductive capability of male animals.

Notably, testes development and reproductive function are tightly regulated by specific signaling pathways, among which steroidogenesis and spermatogenesis regulation pathways are of core importance. The steroidogenesis pathway is responsible for the synthesis of androgens (e.g., testosterone), which are pivotal for initiating sexual maturation, maintaining libido, and promoting testes growth in male Kazakh horses. The spermatogenesis regulation pathway directly governs the proliferation and differentiation of spermatogonia, as well as the production of mature sperm, thereby determining semen quality (e.g., sperm count and motility) and fertilization capacity—key indicators of breeding value in stallions. For Kazakh horses, a breed facing population decline, enhancing reproductive efficiency through breeding improvement is imperative. Elucidating the regulatory mechanisms of these two pathways will not only clarify the molecular basis of superior reproductive traits (e.g., early sexual maturity, high semen quality) but also provide actionable targets for marker-assisted selection. Specifically, key genes within these pathways can be used as molecular markers to screen elite breeding stallions at an early stage, optimize breeding strategies, and stably inherit the breed’s excellent traits. This will ultimately contribute to the conservation and sustainable development of the Kazakh horse population.

Several studies have reported transcriptomic analyses of testicular tissues in various animal species, including cattle [14], sheep [10], pigs [15], and camels [16]. La et al. [17] performed transcriptome sequencing on yak testes tissue at different developmental stages, identifying 30, 23, and 277 common differentially expressed mRNAs (DEmRNAs) at 6, 18, and 30 months of age, respectively. Enrichment analysis revealed that these DEmRNAs were primarily involved in processes such as gonadal embryonic development, cell differentiation, and spermatogenesis. Xi et al. [18] demonstrated that numerous DEGs—including *GATA4*, *GATA6*, *SMAD4*, *SOX9*, *YAP1*, *ITGB1*, and *MAPK1*—were identified in testicular tissues from sheep at different ages. These DEGs showed significant enrichment in the MAPK signaling pathway, ECM-receptor interactions, the PI3K-Akt signaling pathway, and the FOXO signaling pathway. To investigate the molecular mechanisms of testicular development in Kazakh horses, this study employed transcriptomics for the sequencing and analysis of testicular samples. It revealed the biological changes in Kazakh horse testicular development, identified DEGs and associated signaling pathways in testicular tissues before and after sexual maturity, and elucidated key genes and biological processes regulating testicular development. This research provides a foundation for genetic breeding and reproductive performance enhancement in equids.

## 2. Materials and Methods

### 2.1. Experimental Animals

This experiment was conducted in February 2025 in the Tacheng Prefecture of Xinjiang. Eight Kazakh horses raised under identical conditions were selected as experimental animals and divided into two groups: pre-sexual maturity (Group G1, *n* = 4; 1 year old) and post-sexual maturity (Group G3, *n* = 4; 3 years old). Each horse underwent surgical partial castration on the left side, with testicular tissue rapidly preserved in liquid nitrogen and 4% paraformaldehyde fixative for subsequent use. All horses were housed in separate compartments within the same barn, fed a uniform diet of high-quality dry alfalfa and corn kernels, and provided with ample drinking water.

### 2.2. Microscopic Morphological Examination

Testicular tissue was fixed in 10% neutral buffered formalin, dehydrated using an analytical alcohol series, cleared with xylene, embedded in paraffin, and sectioned at 4–5 µm thickness. Tissue sections were prepared and stained with H&E, then photographed under an optical microscope (Eclipse E100 Nikon, Tokyo, Japan).

### 2.3. Transcriptome Sequencing

Total RNA was extracted from testicular tissues using the Trizol™ Kit (Invitrogen, Carlsbad, CA, USA) following the manufacturer’s standard protocol. RNA quality was strictly assessed through dual verification: (1) RNase-free agarose gel electrophoresis (1.5%) was used to visualize the integrity of 28S and 18S rRNA bands, with qualified samples showing clear, non-degraded bands without smearing; (2) an Agilent 2100 Bioanalyzer (Agilent Technologies, Palo Alto, CA, USA) was employed to determine the RNA Integrity Number (RIN ≥ 8.0) and A260/A280 ratio (1.8–2.0), ensuring only high-quality RNA was used for subsequent experiments. mRNA was enriched from total RNA using oligo(dT) magnetic beads (NEB, Ipswich, MA, USA) that specifically bind the poly(A) tail of mRNA, followed by two rounds of purification to eliminate rRNA and other non-coding RNAs. The enriched mRNA was fragmented into 200–300 bp fragments using NEBNext First Strand Synthesis Reaction Buffer (5×) at 94 °C for 8 min.

First-strand cDNA was synthesized with random hexamer primers and M-MuLV Reverse Transcriptase (NEB #M0253), and second-strand cDNA was generated using DNA Polymerase I (NEB #M0209) and RNase H (NEB #M0297). The double-stranded cDNA was purified with AMPure XP Magnetic Beads (Beckman Coulter, Brea, CA, USA), then subjected to end repair, A-tailing, and adapter ligation using the NEBNext Ultra RNA Library Preparation Kit for Illumina (NEB #7530, New England Biolabs, Ipswich, MA, USA). After ligation, the products were purified again with 0.8× volume of AMPure XP Magnetic Beads to select target fragments. PCR amplification was performed using Phusion High-Fidelity DNA Polymerase (NEB #M0530) with the following program: initial denaturation at 98 °C for 30 s; 12 cycles of denaturation at 98 °C for 10 s, annealing at 60 °C for 30 s, and extension at 72 °C for 30 s; final extension at 72 °C for 5 min. The amplified libraries were quantified using a Qubit 4 Fluorometer (Thermo Fisher Scientific, Waltham, MA, USA) and validated for insert size (200–400 bp) via Agilent 2100 Bioanalyzer. Qualified libraries were pooled in equimolar amounts and sequenced on the Illumina Novaseq 6000 platform (Gene Denovo Biotechnology Co., Ltd., Guangzhou, China) with paired-end 150 bp (PE150) mode, generating approximately 8 G of clean data per sample.

### 2.4. Data Quality Control and Verification

Raw data in FASTQ format obtained from sequencing platforms contains adapter sequences and low-quality reads, making it unsuitable for direct alignment analysis. To ensure analytical quality, raw reads were processed to generate clean reads. Raw reads were processed using Fastp (v0.23.2) to remove Illumina adapter sequences and low-quality bases. Simultaneously, quality metrics including Q20, Q30, and GC content were calculated. Clean reads were aligned to the Equus caballus reference genome (EquCab3.0) using HISAT2 (v2.2.1), with gene annotation from Ensembl release 110.

Differential expression analysis of genes between the G1 and G3 groups was performed using the DESeq2 package (v1.34.0) in R software (v4.3.1). Dispersion was estimated using the default method of DESeq2, which employs a shrinkage approach to stabilize variance across genes. Differentially expressed genes (DEGs) were identified with the criteria of |log_2_ fold change| ≥ 1.5 and adjusted *p*-value (Padj) ≤ 0.05, with *q* ≤ 1.00 used to correct *p*-value calculations. The Padj was calculated using the Benjamini and Hochberg (BH) approach to correct for multiple testing, with *q* ≤ 1.00 set as the false discovery rate (FDR) control threshold.

Gene expression levels were quantified as Fragments Per Kilobase of transcript per Million mapped fragments (FPKM) using Cufflinks (v2.2.1). The parameters were set as default, with FPKM calculated based on the length of the transcript and the number of mapped fragments. FPKM was selected as the quantification index because it eliminates the influence of transcript length and sequencing depth, enabling accurate comparison of gene expression levels between different genes and samples. Additionally, raw read counts for each gene were generated using HTSeq (v0.13.5) for subsequent differential expression analysis.

### 2.5. Inter-Sample Correlation Analysis

Principal component analysis (PCA) was performed using R software version 4.3.1 based on gene expression data. By employing a dimension reduction method, the distance relationships among samples were analyzed to evaluate differences in expression patterns between the G1 and G3 groups as well as intra-group consistency. Correlation analysis was conducted to assess gene expression relationships among samples, and heatmaps were used to visualize the resulting correlation coefficients, illustrating pairwise correlations between samples.

### 2.6. GO and KEGG Enrichment Analysis

Enrichment analysis was performed for differentially expressed mRNAs, with GO and KEGG pathway analysis conducted for annotation. KOBAS 3.0 software was used to test the statistical enrichment of DEGs in KEGG pathways, while GOseq software performed GO functional analysis. The significance threshold for enrichment was set at a *p*-value < 0.05.

### 2.7. PPI Network Construction and Key Gene Screening

Predict the interactions of proteins encoded by differentially expressed genes using the STRING database (https://string-db.org/) accessed on 5 November 2025, screen interactions with a confidence score > 0.7 to construct a PPI network, and visualize it using Cytoscape v3.9.0 software.

### 2.8. RT-qPCR Validation

Glyceraldehyde-3-phosphate dehydrogenase (GAPDH) and β-actin (ACTB) are chosen as internal control genes for RT-qPCR validation. Extract total RNA from the testes sample, take a grinding tube, add 1 mL of RNA extraction solution, add three 3 mm grinding beads, and pre-cool on ice. Take 5–20 mg of tissue and add it to the grinding tube. Grind well until there are no visible tissue blocks. Centrifuge at 12,000 rpm for 10 min at 4 °C to take the supernatant. Add 100 μL of chloroform substitute, invert the centrifuge tube for 15 s, mix well, and let stand for 3 min. Centrifuge at 12,000 rpm for 10 min at 4 °C. Transfer 400 μL of the supernatant to a new centrifuge tube, add 550 μL of isopropanol, and mix by inverting. Leave at −20 °C for 15 min. Centrifuge at 12,000 rpm at 4 °C for 10 min, and the white precipitate at the bottom of the tube is RNA. Aspirate the liquid, add 1 mL of 75% ethanol, and mix to wash the pellet. Centrifuge at 12,000 rpm for 5 min at 4 °C. Repeat steps 10–11 once. Suck the liquid clean, put the centrifuge tube on the clean table, and blow for 3–5 min. Add 15 μL of RNA lysis solution to dissolve RNA. Use Nanodrop 2000 to detect RNA concentration and purity: after the instrument blank is zeroed, take 2.5 μL of the RNA solution to be tested on the detection base, put down the sample arm, and use the software on the computer to start the absorbance value detection. Dilute the RNA that is too high in an appropriate ratio to a final concentration of 200 ng/μL. Reverse transcription of total RNA into cDNA. Gently mix the reverse transcription reaction (20 μL reaction set, reverse transcription kit catalog number G3337) and centrifuge. Set up a reverse transcription program, and complete reverse transcription on a common PCR instrument for RT-qPCR primer information. Take 0.1 mL of the PCR reaction plate and prepare the reaction system as follows, with 3 tubes of each reverse transcript product. After spotting the sample, complete the sealing film with PCR sealing film and sealing instrument, and carry out centrifugation with a microplate centrifuge. Perform PCR amplification on a real-time PCR instrument. All samples were subjected to 3 technical replicates. All lab equipment consumables were obtained as shown in (Appendix A).ΔΔCT method: A = CT (target gene, sample to be tested) − CT (internal standard gene, sample to be tested)B = CT (target gene, control sample) − CT (internal standard gene, control sample)K = A − BExpression fold = 2 − K

Data normalization was performed using the 2^−ΔΔCT^ method as described above. Statistical analysis of RT-qPCR results was conducted using GraphPad Prism 9 software. An independent samples *t*-test was used to compare the ΔCt values of target genes between the G1 (pre-sexual maturity) and G3 (post-sexual maturity) groups. Differences were considered statistically significant at *p* < 0.05.

## 3. Results and Analysis

### 3.1. Morphological Observations of Testicular Tissue in Kazakh Horses

As shown in Figure 1A,B, during the G1 stage, the number of interstitial cells is relatively low, lumens have not yet formed, and the basement membrane is thin. As depicted in Figure 1C,D, in the G3 stage, interstitial cells are densely packed with abundant cytoplasm, and early spermatocytes and a small number of spermatozoa are visible. This indicates that both the development of the luminal basement membrane and changes in interstitial cell numbers during testicular development reflect the progressive maturation of testicular function. These morphological changes clearly indicate the structural maturation of Kazakh horse testes from pre-sexual maturity to post-sexual maturity. To further elucidate the molecular mechanisms underlying this developmental process, we performed transcriptome sequencing and subsequent bioinformatics analysis on testicular tissues from both groups.

### 3.2. RNA Sequencing Data Analysis

This study generated a total of eight cDNA libraries. As shown in Table 1, the testes transcriptome yielded approximately 650 million (mean 81,877,347.5) high-quality reads; GC content in testes transcriptome sequencing ranged from 49.18% to 51.29%; Q20 percentage ranged from 98.52% to 98.67%; and Q30 read coverage ranged from 93.82% to 94.35%. Furthermore, over 99.25% of the cleaned data from testes tissue aligned with the reference genome.

### 3.3. Inter-Sample Expression Pattern Analysis of Testicular Tissue Samples from Kazakh Horses

To evaluate the consistency within groups and differences between groups, we performed principal component analysis (PCA), Venn diagram analysis, and correlation analysis based on gene expression data. The correlation analysis of testicular tissue samples from 1-year-old and 3-year-old Kazakh horses revealed that, as shown in Figure 2A, PCA1 was the primary coordinate in the principal component analysis (PCA) of Groups G1 and G3, representing a contribution rate of 96.5%. PCA2 was the second principal coordinate, accounting for 2.5% of the variance. Intra-group clustering was compact while inter-group clustering was loose, indicating significant differences between the two groups. As shown in Figure 2B, the G1 group contained 13,733 genes, while the G3 group contained 13,045 genes, with 12,884 genes shared between the two groups. As shown in the results of Figure 2C,D, due to minor variations in individual expression levels and overall consistent expression patterns across samples, the correlation between samples within each group also exhibits a similar stable trend.

### 3.4. Pattern Analysis of Differentially Expressed Genes of Testicular Tissue Samples from Kazakh Horses

Analysis of differentially expressed genes in Kazakh horse testicular tissue revealed that, as shown in Figure 3A,B, a total of 3054 DEGs were identified between the G1 and G3 groups, including CABS1, RPL10, PGAM2, TMSB4X, and CYP17A1. Among these, 402 genes showed upregulation and 2652 genes exhibited downregulation (see Appendix A). The clustering analysis results between samples are shown in Figure 3C. The DEGs in the testicular tissue samples demonstrated high reproducibility across groups, revealing significant differences between them. Differential expression analysis was conducted. The results of the selection of top DEGs are shown in the Appendix A (see Appendix A).

### 3.5. GO Functional Annotation and KEGG Enrichment Analysis of Differentially Expressed Genes in Kazakh Horse Testicular Tissue

To analyze the functions of DEGs across different testicular tissue samples, GO and KEGG enrichment analyses were performed on the DEGs. The functional enrichment analysis of DEGs was categorized into three groups: biological processes (BP), molecular functions (MF), and cellular components (CC).

As shown in Figure 4A, GO annotation results for the G1 and G3 groups indicate that DEGs were primarily enriched in annotation categories including Cellular process (BP), Single-organism process (BP), Biological regulation (BP), Binding (MF), Catalytic activity (MF), Molecular transducer activity (MF), Cell (CC), Cell part (CC), and Membrane (CC). As shown in Figure 4B, KEGG enrichment analysis of G1 and G3 groups indicates that DEGs are predominantly enriched in signaling pathways, including Focal adhesion, Pathways in cancer, and the PI3K-Akt signaling pathway. (See Appendix A)

### 3.6. Protein–Protein Interactions

As shown in Figure 5, to investigate the molecular mechanisms underlying testicular development in Kazakh horses, a protein–protein interaction (PPI) network was constructed based on differentially expressed mRNAs to analyze their interaction network. In the G1 and G3 groups, nine key genes were identified: *CYP11A1*, *SOX9*, *HAO2*, *RPS4X*, *TCEA3*, *SULT1E1*, *DCN*, *GATA6*, and *CREBP*.

### 3.7. RT-qPCR Results Validation

To validate the accuracy of transcriptomic sequencing data, this study randomly selected differentially expressed genes *CYP17A1*, *CYP19A1*, *RPL14*, *TMSB4X*, *PGAM2*, *IZUMO4*, *RPS4X*, and *CABS1* for RT-qPCR verification. As shown in Figure 6, all the gene expression trends were consistent in both RT-qPCR and RNA sequencing results, indicating that the sequenced data and expression levels identified in this study are reliable and can be used for subsequent analyses.

## 4. Discussion

The testes are one of the primary reproductive organs in male animals, functioning to produce sperm and secrete androgens [19]. Composed of seminiferous tubules and interstitial tissue, the testes feature convoluted tubular structures lined internally by a layer of spermatogenic cells. These cells reside within the seminiferous tubules, undergoing a series of developmental and differentiation processes to ultimately generate sperm [20]. The testicular interstitium contains supportive structures such as blood vessels and connective tissue, supplying the testes with essential nutrients and oxygen [21]. Thus, normal testicular development is crucial for maintaining male reproductive health and fertility. This study identified DEGs by comparing testicular tissues from sexually mature and immature Kazakh horses. Among these, genes such as *CABS1*, *RPL10*, *PGAM2*, *TMSB4X*, and *CYP17A1* may be involved in biological processes including testicular development and spermatogenesis.

The core function of *CABS1* is calcium signaling regulation, acting as a calcium-binding protein to maintain intracellular calcium homeostasis and calcium-dependent signaling pathways [22]. Zhang et al. [23] demonstrated that *CABS1* is a novel spermatocyte-specific protein whose genetic deletion does not affect testes and epididymis development, yet results in impaired sperm tail structure and reduced fertility in male mice. In recent years, *CABS1* has been found to be highly expressed or enriched in elongated spermatocytes in human, pig, and mouse testes [24,25,26]. Zhang et al. [27] identified *CABS1* as a fertility-related gene through whole-genome sequencing of the Wanbei pig. This gene is highly expressed in testicular germ cells, regulating spermatogonial meiosis, spermatogenesis, and sperm motility by controlling intracellular calcium levels. Its abnormal expression may cause sperm morphological abnormalities or reduced fertilization capacity. As a highly complex organ, the testes undergo developmental changes from the embryonic stage to adulthood. The development of germ cells producing sperm is regulated by the surrounding somatic cells. Salehi et al. [28] demonstrated through single-cell RNA sequencing that both somatic and germ cell populations exhibit significant heterogeneity. The highest diversity was observed in supporting cells, myoid cells, spermatogonia, spermatocytes, and spermatogonial germ cells. Key somatic genes identified include *RPL10*, *RPL39*, *RPL13A*, and *FTH1*. These genes exert high influence within the weighted gene co-expression network of the testes transcriptomic atlas and are associated with male infertility. *RPL10* participates in protein synthesis and translational regulation while also functioning as a transcription factor to regulate downstream gene expression. Testes development and spermatogenesis demand substantial de novo protein synthesis, including germ cell-specific proteins and structural proteins. By safeguarding ribosomal function, *RPL10* provides protein synthesis support for spermatogonial proliferation, spermatocyte differentiation, and spermatogenesis. Its expression level directly correlates with testicular tissue metabolic activity and spermatogenic efficiency [29]. *CYP17A1* catalyzes steroid hormone synthesis, playing a crucial role in the biosynthesis of androgens and glucocorticoids [30]. Primarily expressed in testicular interstitial cells, it regulates seminiferous tubule development, supports Sertoli cell function, and regulates germ cell differentiation by synthesizing testosterone. Its expression level directly influences testicular developmental maturity, and abnormal expression may lead to delayed testicular development, spermatogenesis disorders, or male infertility [31]. Lea et al. [32] conducted histomorphological analysis of fetal sheep testes. Results showed that at day 140 of gestation, the number of Sertoli cells remained constant throughout pregnancy, while *CYP17A1* and *CYP11A1* staining in interstitial cells decreased. However, extracting fetuses at day 80 reduced both Sertoli cell numbers and *CYP17A1* immunoreactivity in interstitial cells. Thus, *CYP17A1* is a key regulatory gene for male reproductive system development, and its mutations or abnormal expression are closely associated with reproductive dysfunction in animals.

*GSTA3* belongs to the glutathione S-transferase family, and its core function is to eliminate reactive oxygen species (ROS) and detoxify exogenous harmful substances. During spermatogenesis, the mitochondria of reproductive cells are highly active, which leads to the production of a large amount of ROS. Excessive ROS can cause damage to sperm DNA and a decrease in sperm motility [33]. In mature testes, *GSTA3* is upregulated, which can enhance the antioxidant capacity and thereby protect sperm quality. *GLIPR1L1* is a gene specifically expressed in reproductive cells and regulates the formation of sperm acrosomes and the combination of sperm and egg [34]. In immature testes, only spermatogonial cells are present, and no acrosome-related functions are required; in mature testes, spermatogonial cells differentiate into sperm cells, and the formation of acrosomes (containing hydrolases for penetrating the zona pellucida of the egg cell) is required. The upregulation of *GLIPR1L1* indicates that sperm have transformed from immature cells to mature sperm with high fertilization ability, and is a direct molecular marker of testicular function maturation. *IGF2* and *IGFBP3* are growth regulatory genes during the embryonic stage. *IGF2* is the core growth factor during the embryonic stage, promoting the proliferation of immature testicular tissue. *IGFBP3* is the binding protein that regulates the activity of *IGF2*. In immature testes, *IGF2* drives the dilation of the spermatogenic tubules and the proliferation of stromal cells to construct the tissue framework; after maturation, the testis volume stabilizes, and excessive proliferation can lead to tissue disorder (such as crowded spermatogenic tubules) [35,36,37]. The downregulation of their expression can completely terminate the growth signal, causing the testes to shift from volume expansion to function maintenance.

This study performed GO functional annotation and KEGG enrichment analysis on the differentially expressed genes. Results indicate that most differentially expressed mRNAs participate in multiple biological processes, including immunity, growth, metabolism, development, and reproduction. Therefore, these DEGs play a crucial regulatory role in the development of Kazakh horse testes before and after sexual maturity. KEGG enrichment analysis revealed that the differential mRNAs were primarily enriched in pathways such as Focal adhesion, Pathways in cancer, and the PI3K-Akt signaling pathway. Focal adhesion constitutes a dynamic structural link between cells and the extracellular matrix (ECM), composed of integrin transmembrane receptors, cytoskeletal proteins (e.g., actin), and signaling proteins. It functions as a mechanical anchor and signaling hub. Fu et al. [38] conducted transcriptome analysis on testicular tissues from Hu sheep and Tibetan sheep, revealing 466 DEGs significantly enriched in pathways related to testes development and spermatogenesis, including Focal adhesion, cAMP signaling pathway, and P53 signaling pathway. Among these, *COL1A1* and *COL1A2* regulate the adhesion of spermatogonia and spermatocytes to the basement membrane, as well as the separation and migration of spermatocytes and spermatozoa toward the lumen during late stages, thereby promoting spermatogenesis and release. Wang et al. [39] demonstrated that transcriptomic sequencing analysis of primary cultured yak Leydig cells (LCs) and Sertoli cells (SCs) revealed DEGs predominantly enriched in signaling pathways, including focal adhesion, Rap1/MAPK signaling pathway, and steroid biosynthesis. Differences between LCs and SCs were primarily reflected in steroid hormone synthesis, cell proliferation and metabolism, and dynamic regulation of the blood–testes barrier (BTB). Thus, maintaining normal BTB physiological function can positively influence normal testicular development in sheep. The PI3K-Akt signaling pathway is one of the most critical regulatory mechanisms in animal cells, primarily governing cell proliferation, survival, and anti-apoptotic responses [40]. Its activation promotes proliferation and anti-apoptotic effects in immature supporting cells and spermatogenic cells. Furthermore, in mature supporting cells, this pathway disrupts BTB architecture by regulating protein synthesis and the cytoskeleton of supporting cells. These actions directly or indirectly maintain and promote spermatogenesis in male testes [41]. Wang et al. [42] performed single-cell transcriptomic sequencing analysis of yak testes tissue, identifying six somatic cell types and various germ cell lineages, including spermatogonial stem cells, spermatogonia, early spermatocytes, late spermatocytes, and spermatids. Functional enrichment analysis revealed that genes expressed in yak testicular somatic cells were significantly enriched in the cAMP signaling pathway, PI3K-Akt signaling pathway, MAPK signaling pathway, and ECM receptor interactions. During testicular development from juvenile to adult stages, the proliferation and differentiation of supporting cells and interstitial cells, as well as the structural formation of seminiferous tubules, all depend on the activation of the PI3K-Akt signaling pathway [43]. This pathway likely promotes supporting cell proliferation to provide nutritional support for germ cells while suppressing abnormal apoptosis in testicular tissue. This maintains cellular homeostasis and ensures the normal developmental transition of testicular tissue. The renewal and proliferation of spermatogonia, initiation of meiosis in spermatocytes, and morphological remodeling of spermatids all require regulation of downstream gene expression by the PI3K-Akt signaling pathway. Insufficient pathway activation impairs germ cell proliferation or increases apoptosis, leading to oligospermia or azoospermia. Conversely, excessive activation may cause abnormal germ cell proliferation, adversely affecting sperm quality [44].

## 5. Conclusions

This study conducted transcriptome sequencing analysis on the testicular tissues of Kazakh horses before and after sexual maturity. The results showed that core differentially expressed genes (DEGs), such as *CABS1*, *RPL10*, *PGAM2*, *TMSB4X,* and *CYP17A1*, were significantly enriched in pathways related to cell development, cell proliferation, and the secretion of sex hormones, which are involved in biological processes such as focal adhesion, pathways in cancer, and the PI3K-Akt signaling pathway, and play positive roles. The accuracy of the data results was verified by RT-qPCR. Therefore, the selected core DEGs and key signaling pathways regulate the development of Kazakh horse testicular tissues and promote spermogenesis, providing a reliable basis for reproductive breeding of equine species and the protection and improvement of the Kazakh horse population.

## Figures and Tables

**Figure 1 biology-15-00100-f001:**
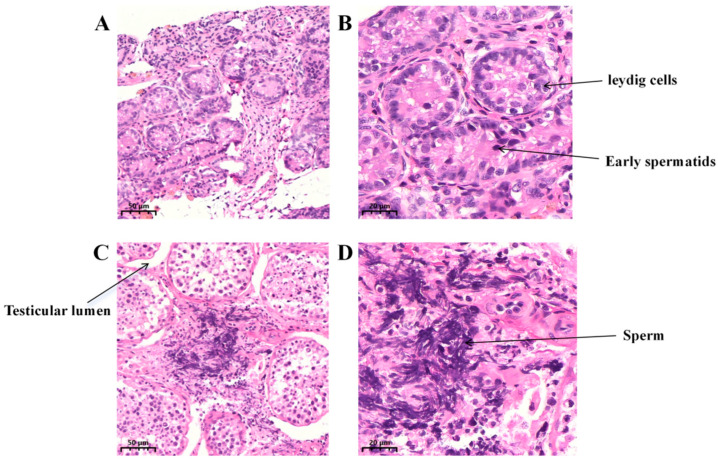
Histological examination of testicular tissue in groups G-1 and G-3. (**A**) G1 group testicular tissue section at 20× magnification; (**B**) G1 group testicular tissue section at 50× magnification; (**C**) G3 group testicular tissue section at 20× magnification; (**D**) G3 group testicular tissue section at 50× magnification.

**Figure 2 biology-15-00100-f002:**
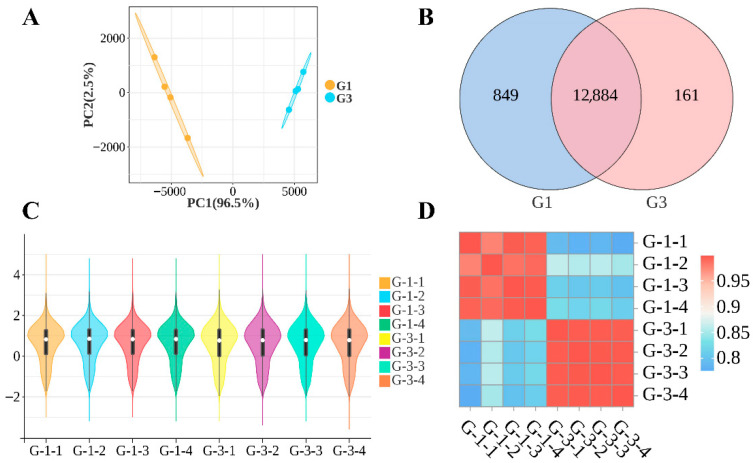
Correlation analysis chart between G-1 and G-3 groups (**A**) PCA plot of samples from group G1 and group G3; (**B**) Venn diagram for groups G1 and G3; (**C**) Violin plot of samples between groups G1 and G3; (**D**) Correlation heatmap between group G1 and group G3.

**Figure 3 biology-15-00100-f003:**
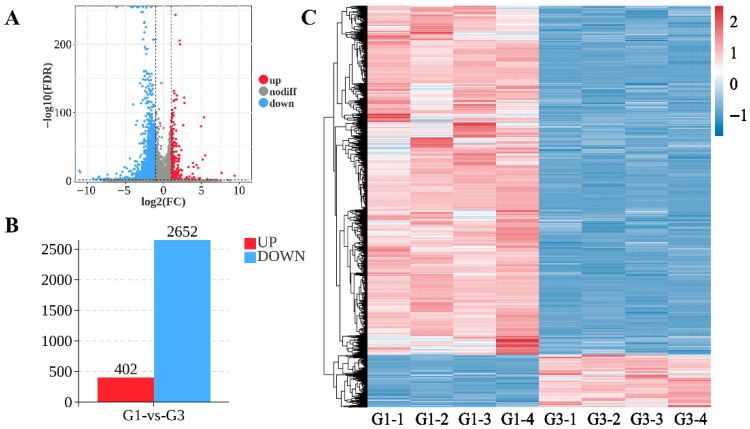
Differential expression analysis diagram for groups G-1 and G-3 (**A**) Volcano plot of differentially expressed genes between group G1 and group G3; (**B**) Bar chart of differentially expressed genes between group G1 and group G3; (**C**) Heatmap of differentially expressed genes between group G1 and group G3. Note: In (**A**,**B**), ‘up’ and ‘down’ represent gene expression upregulation and downregulation in testicular tissue, respectively. Upregulation: (log_2_FC > 1 and adjusted *p* < 0.05), downregulation: (log_2_FC < −1 and adjusted *p* < 0.05). In (**C**), the horizontal axis represents individual samples, the vertical axis represents expression levels, and the transition from blue to red indicates progressively increasing upregulation.

**Figure 4 biology-15-00100-f004:**
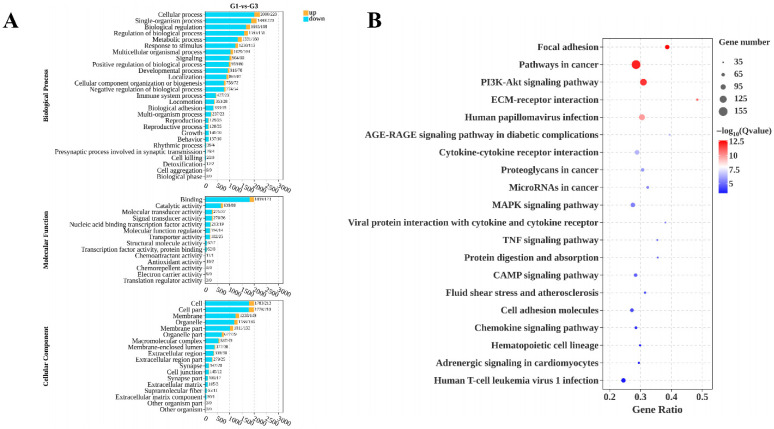
GO and KEGG enrichment analysis diagram for groups G-1 and G-3 (**A**) GO annotation entry diagram for group G1 and group G3; (**B**) KEGG enrichment analysis diagram for groups G1 and G3. Note: In (**A**), the horizontal axis represents each DEG, while the vertical axes denote biological processes (BP), molecular functions (MF), and cellular components (CC). (**B**) depicts the top 20 pathways with the lowest Q-values. The vertical axis displays pathway names, and the horizontal axis shows gene ratios. Count values increase from left to right, with blue to red indicating progressively higher Q-values.

**Figure 5 biology-15-00100-f005:**
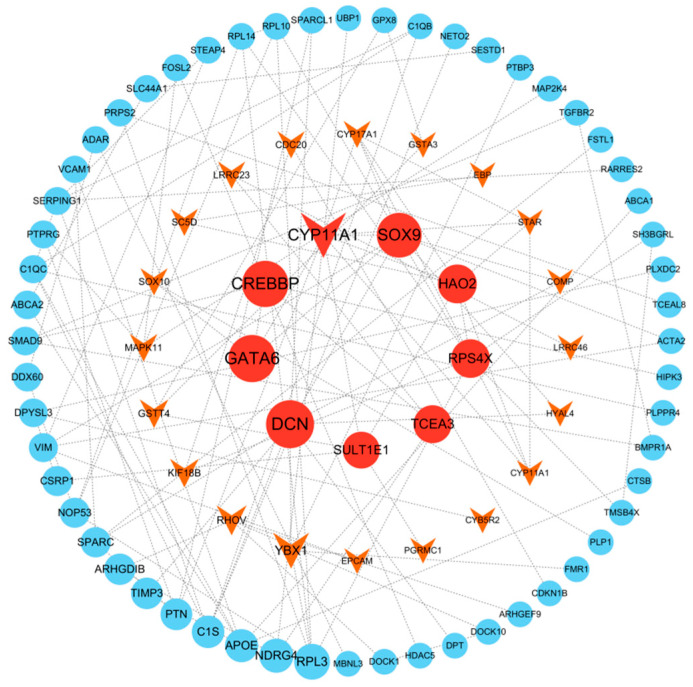
PPI protein interaction network map. Note: Triangles represent upregulated genes, circles represent downregulated genes. The screening criteria for key differentially expressed genes are: |log_2_(fold change)| ≥ 1.5 and adjusted *p*-value ≤ 0.05.

**Figure 6 biology-15-00100-f006:**
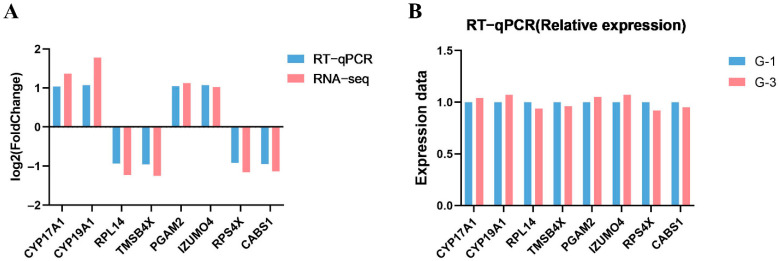
Validation of differentially expressed genes by RT-qPCR. (**A**) Log_2_ fold-change comparison between RNA sequencing and RT-qPCR for differentially expressed genes; (**B**) Relative expression of differentially expressed genes by RT-qPCR.

**Table 1 biology-15-00100-t001:** Overall detection of mRNA sequencing data.

Samples	Raw Data	Clean Data	Q20	Q30	GC Content	Mapped Reads
G-1-1	84,567,774	84,258,310 (99.63%)	98.60%	94.17%	50.15%	83,502,734 (99.10%)
G-1-2	80,621,274	80,320,920 (99.63%)	98.67%	94.33%	49.18%	79,767,150 (99.31%)
G-1-3	81,816,144	81,557,174 (99.68%)	98.62%	94.14%	49.94%	80,865,568 (99.15%)
G-1-4	85,756,968	85,413,322 (99.60%)	98.62%	94.35%	49.74%	84,747,352 (99.22%)
G-3-1	82,110,868	81,803,308 (99.63%)	98.54%	93.93%	51.28%	81,199,492 (99.26%)
G-3-2	82,664,624	82,343,798 (99.61%)	98.61%	94.22%	51.05%	81,829,900 (99.38%)
G-3-3	73,231,020	72,926,266 (99.58%)	98.52%	93.83%	51.29%	72,412,834 (99.30%)
G-3-4	84,250,108	83,918,332 (99.61%)	98.55%	93.82%	50.93%	83,319,694 (99.29%)

Note: Samples: Sample name; Raw Data: Original sequencing data; Clean Data: Filter the data; Q20: Proportion of bases with quality value greater than or equal to 20; Q30: Proportion of bases with quality value greater than or equal to 30; GC content: Calculate the percentage of the total number of bases G and C to the total number of bases; Mapped reads: Comparing reads to the genome.

## Data Availability

The data presented in this study are openly available in BioProject with reference number PRJNA1376256.

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
