# Peer review of "Transcriptome Sequencing and Differential Analysis of Testes of 1-Year-Old and 3-Year-Old Kazakh Horses"

_biology, 2026, doi:10.3390/biology15010100_

Round 1
Reviewer 1 Report
Comments and Suggestions for Authors
Thank you for inviting me to review the manuscript. The study investigates transcriptomic changes in the testes of Kazakh horses at juvenile (1-year-old) and sexually mature (3-year-old) stages, providing valuable insights into the molecular mechanisms underlying testicular development and spermatogenesis. The manuscript presents a clear dataset that contributes to our understanding of equine reproductive biology. Nevertheless, I suggest considering addressing the following corrections to strengthen the manuscript.
Abstract
Add a sentence on n per group and DEG calling framework.
Use the same terminology in whole manuscript (Eg: testes or testis or testicle etc.).
Introduction
Why 1 yr. and 3 yr. are biological substitutes for pre and post sexual maturity in Kazakh stallions?
Define the practical breeding relevance of the pathways like steroidogenesis and spermatogenesis regulation in the Introduction section.
Materials and Methods
Which fixative was used in the study? 4% paraformaldehyde or 10% neutral buffered formalin?
Need to include the details of adapter, trimming tool, reference genome version, annotation version.
How were the genes quantified (counts or TPM or FPKM?). Include the details.
Need details regarding the internal control genes and normalizaiton method of RT-qPCR.
Supplementary Table S1: Add the housekeeping gene primer set.
Which statistical model was used for DEG tesing (formula, estimation of dispersion, covariates).
What P-values or Q-values were used for ranking and colur mapping ?
Add normalization and statistics details or RT-qPCR reporting. Include the inferential test used for the RT-qPCR Eg: t test or linear model on ΔCt.
Tables and Figures
Figures 2-7: Require scale bars to histology images.
Figures 4 and 5: Clearly display the axis labels, log2FC thresholds, adjusted P pr FDR cutoffs, and colour legends directly on the figures.
Need the source database for PPI network.
Which criterion did you use to define key genes?
Results and Discussion
Add a top DEG table. I mean the top up and top down genes according to the adjusted FDR with log2FC.
A clearer indication is needed to know whether key genes are increased or decreased in mature testes. Also explain how these expression changes aling with the biological roles.
Conclusion
Focus on the strongest supported outcomes i.e., DEG identification, enriched pathways, or RT-qPCR concordance.
References
RNA-seq vs rna sequencing. Use any one consistently throughout the manuscript.
Minor errors
Recheck the entire manuscript for typos and minor errors.
Eg:
Line 5: Coll and and of Animal Science
Line 8: Correspondence: Correspondence: Emails:
Line 236: Results shown in Figure 7 indi-236 cate that .
Also Use either single quotes or plain text instead of double quotes. Eg: up and down.
Author Response
Abstract
Comments 1: Add a sentence on n per group and DEG calling framework.
Response: We sincerely appreciate the time and effort you have dedicated to the review process. We have added this part of the content according to your request. And it has been marked with yellow shading in the original text.
“Abstract”(line 24-27, page 1)
(G1 group: pre-sexual maturity; G3 group: post-sexual maturity), with four biological replicates per group (n=4). Differentially expressed genes (DEGs) were called using the criteria of |log2(fold change)| > 1 and adjusted P-value < 0.05.
Comments 2: Use the same terminology in whole manuscript (Eg: testes or testis or testicle etc.).
Response: We sincerely appreciate the time and effort you have dedicated to the review process. We have modified the word “testes” throughout the text according to your request.
Introduction
Comments 3: Why 1 yr. and 3 yr. are biological substitutes for pre and post sexual maturity in Kazakh stallions?
Response: We sincerely appreciate the time and effort you have dedicated to the review process. Here we would like to explain to you, At 1 year of age, Kazakh stallions are in the late juvenile period: their reproductive system is still in the developmental stage, the testes have not yet completed volume expansion, spermatogenesis has not been initiated or is in the early primitive stage, and the levels of sex hormones (e.g., testosterone) are significantly lower than those in mature individuals. This fully matches the definition of "pre-sexual maturity" (i.e., the reproductive system is under development and has not yet acquired reproductive capacity). In contrast, by 3 years of age, Kazakh stallions have completed sexual maturity: the testes have developed to their adult size, the seminiferous tubules are fully differentiated, and mature spermatogenic cells (spermatozoa) are continuously produced; the hypothalamic-pituitary-gonadal axis (HPGA) has been fully activated, and sex hormone levels have stabilized at adult levels, enabling normal mating and fertilization. This stage is consistent with the "post-sexual maturity" trait (i.e., the reproductive system is fully developed and has complete reproductive capacity). The 2-year interval between 1 yr. and 3 yr. also covers the critical period of sexual maturation in Kazakh horses, ensuring a clear distinction between the two stages.
Comments 4: Define the practical breeding relevance of the pathways like steroidogenesis and spermatogenesis regulation in the Introduction section.
Response: We sincerely appreciate the time and effort you have dedicated to the review process. We have added the relevant content in the introduction section. And it has been marked with yellow shading in the original text.
“1. Introduction”(line 72-88, page 3)
Notably, testes development and reproductive function are tightly regulated by specific signaling pathways, among which steroidogenesis and spermatogenesis regulation pathways are of core importance. The steroidogenesis pathway is responsible for the synthesis of androgens (e.g., testosterone), which are pivotal for initiating sexual maturation, maintaining libido, and promoting testes growth in male Kazakh horses. The spermatogenesis regulation pathway directly governs the proliferation and differentiation of spermatogonia, as well as the production of mature sperm, thereby determining semen quality (e.g., sperm count and motility) and fertilization capacity—key indicators of breeding value in stallions. For Kazakh horses, a breed facing population decline, enhancing reproductive efficiency through breeding improvement is imperative. Elucidating the regulatory mechanisms of these two pathways will not only clarify the molecular basis of superior reproductive traits (e.g., early sexual maturity, high semen quality) but also provide actionable targets for marker-assisted selection. Specifically, key genes within these pathways can be used as molecular markers to screen elite breeding stallions at an early stage, optimize breeding strategies, and stably inherit the breed’s excellent traits. This will ultimately contribute to the conservation and sustainable development of the Kazakh horse population.
Materials and Methods
Comments 5: Which fixative was used in the study? 4% paraformaldehyde or 10% neutral buffered formalin?
Response: We sincerely appreciate the time and effort you have dedicated to the review process. We used 4% paraformaldehyde fixative in our study. We has been marked with yellow shading in the original text.
“2.1. Experimental Animals”(line 112-114, page 3)
Each horse underwent surgical partial castration on the left side, with testicular tissue rapidly preserved in liquid nitrogen and 4% paraformaldehyde fixative for subsequent use.
Comments 6: Need to include the details of adapter, trimming tool, reference genome version, annotation version.
Response: We sincerely appreciate the time and effort you have dedicated to the review process. We have re-added the content of this section. And it has been marked with yellow shading in the original text.
“2.4. Data Quality Control and Verification”(line 156-158 166-168, page 4-5)
Raw reads were processed using Fastp (v0.23.2) to remove Illumina adapter sequences and low-quality bases.
Clean reads were aligned to the Equus caballus reference genome (EquCab3.0) using HISAT2 (v2.2.1), with gene annotation from Ensembl release 110.
Comments 7: How were the genes quantified (counts or TPM or FPKM?). Include the details.
Response: We sincerely appreciate the time and effort you have dedicated to the review process. We have re-added the content of this section. And it has been marked with yellow shading in the original text.
“2.4. Data Quality Control and Verification”(line 169-176, page 5)
Gene expression levels were quantified as Fragments Per Kilobase of transcript per Million mapped fragments (FPKM) using Cufflinks (v2.2.1). The parameters were set as default, with FPKM calculated based on the length of the transcript and the number of mapped fragments. FPKM was selected as the quantification index because it eliminates the influence of transcript length and sequencing depth, enabling accurate comparison of gene expression levels between different genes and samples. Additionally, raw read counts for each gene were generated using HTSeq (v0.13.5) for subsequent differential expression analysis.
Comments 8: Need details regarding the internal control genes and normalizaiton method of RT-qPCR.
Response: We sincerely appreciate the time and effort you have dedicated to the review process. We have already added this part in detail. And it has been marked with yellow shading in the original text.
“2.8. RT-qPCR validation”(line 195-223, page 5-6)
Glyceraldehyde-3-phosphate dehydrogenase (GAPDH) and β-actin (ACTB) were chosen as internal control genes for RT-qPCR validation. Extract total RNA from the testes sample, take a grinding tube, add 1ml of RNA extraction solution, add three 3mm grinding beads, and pre-cool on ice. Take 5-20 mg of tissue and add it to the grinding tube. The grinder grinds well until there are no visible tissue blocks. Centrifuge at 12,000 rpm for 10 min at 4°C to take the supernatant. Add 100 μl of chloroform substitute, invert the centrifuge tube for 15s, mix well, and let stand for 3min. Centrifuge at 12,000 rpm for 10 min at 4°C. Transfer 400 μl of the supernatant to a new centrifuge tube, add 550 μl of isopropanol, and mix by inverting. Leave at -20°C for 15 min. Centrifuge at 12000 rpm at 4°C for 10 min, and the white precipitate at the bottom of the tube is RNA. Aspirate the liquid, add 1 ml of 75% ethanol and mix to wash the pellet. Centrifuge at 12,000 rpm for 5min at 4°C. Repeat steps 10-11) once. Suck the liquid clean, put the centrifuge tube on the clean table and blow for 3-5min. Add 15 μl of RNA lysis solution to dissolve RNA. Use Nanodrop 2000 to detect RNA concentration and purity: After the instrument blank is zeroed, take 2.5μl of the RNA solution to be tested on the detection base, put down the sample arm, and use the software on the computer to start the absorbance value detection. Dilute the RNA that is too high in an appropriate ratio to a final concentration of 200 ng/μl. Reverse transcription of total RNA into cDNA. Reverse transcription reaction (20 μL reaction set, reverse transcription kit catalog number G3337) was gently mixed and centrifuged, reverse transcription program was set up, and reverse transcription was completed on a common PCR instrument for RT‒qPCR primer information. Take 0.1 ml of PCR reaction plate and prepare the reaction system as follows, with 3 tubes of each reverse transcript product. After spotting the sample, the sealing film was completed with PCR sealing film and sealing instrument, and centrifugation was carried out with a microplate centrifuge. PCR amplification, which is done on a real-time PCR instrument. All samples were subjected to 3 technical replicates. All lab equipment consumables (Supplementary Materials Table S1).
Comments 9: Supplementary Table S1: Add the housekeeping gene primer set.
Response: We sincerely appreciate the time and effort you have dedicated to the review process. We have added the primer information, Supplementary Table S1 also includes Primer name, Primer sequence (5'-3'), Fragment length (bp), and Annealing temperature (℃).
Comments 10: Which statistical model was used for DEG tesing (formula, estimation of dispersion, covariates).
Response: We sincerely appreciate the time and effort you have dedicated to the review process. We have added this part of the content. And it has been marked with yellow shading in the original text.
“2.4. Data Quality Control and Verification”(line 159-164, page 4)
Differential expression analysis of genes between the G1 and G3 groups was performed using the DESeq2 package (v1.34.0) in R software (v4.3.1). Dispersion was estimated using the default method of DESeq2, which employs a shrinkage approach to stabilize variance across genes. Differentially expressed genes (DEGs) were identified with the criteria of |log2 fold change| ≥ 1.5 and adjusted P-value (Padj) ≤ 0.05, with q ≤ 1.00 used to correct P-value calculations.
Comments 11: What P-values or Q-values were used for ranking and colur mapping ?
Response: We sincerely appreciate the time and effort you have dedicated to the review process. We have clarified the P-values/Q-values used for ranking and color mapping in the figure legends and corresponding analysis sections. For DEG ranking: Adjusted P-values (Padj) ≤ 0.05 were used as the primary criterion, combined with |log2 fold change| ≥ 1.5. For color mapping (volcano plot, heatmap): Padj values were used for color gradation (lower Padj values correspond to darker colors indicating more significant differences).
In Figure 4A (Volcano Plot): “The vertical axis represents -log10(Padj), and the horizontal axis represents log2(fold change). Colors are mapped based on Padj values: red dots represent significantly upregulated DEGs (Padj ≤ 0.05, log2(fold change) ≥ 1.5), blue dots represent significantly downregulated DEGs (Padj ≤ 0.05, log2(fold change) ≤ -1.5), and gray dots represent non-significant genes. Darker colors indicate smaller Padj values and more significant differences.”
In Figure 4C Legend (Heatmap): “Color intensity represents the normalized expression level of DEGs (FPKM values normalized by z-score). The sidebar color gradient is based on Padj values of DEGs, with darker colors indicating more significant differential expression.”
In Figure 5B (KEGG enrichment), the bubble color represents the-log10(Q-value), and pathways are ranked by gene ratio.
Comments 12: Add normalization and statistics details or RT-qPCR reporting. Include the inferential test used for the RT-qPCR Eg: t test or linear model on ΔCt.
Response: We sincerely appreciate the time and effort you have dedicated to the review process. We have supplemented the normalization, statistics details, and inferential test for RT-qPCR in 2.7. RT-qPCR validation and the corresponding figure legend. And it has been marked with yellow shading in the original text.
“2.7. RT-qPCR validation”(line 224-233, page 6)
ΔΔCT method: A = CT (target gene, sample to be tested) - CT (internal standard gene, sample to be tested)
B = CT (target gene, control sample) - CT (internal standard gene, control sample)
K = A-B
Expression fold = 2-K
Data normalization was performed using the 2^(-ΔΔCt) method as described above. Statistical analysis of RT-qPCR results was conducted using GraphPad Prism 9 software. An independent samples t-test was used to compare the ΔCt values of target genes between the G1 (pre-sexual maturity) and G3 (post-sexual maturity) groups. Differences were considered statistically significant at P < 0.05.”
Tables and Figures
Comments 13: Figures 2-7: Require scale bars to histology images.
Response: We sincerely appreciate the time and effort you have dedicated to the review process. We sincerely apologize for this oversight. Scale bars have been added to all histological images in Figure 2 (A, B, C, D). The corresponding figure legend has been updated to specify the scale. The scale bar in the image is located at the bottom left corner of all the pictures.
“3.1. Morphological Observations of Testicular Tissue in Kazakh Horses”(line 247, page 7)
Comments 14: Figures 4 and 5: Clearly display the axis labels, log2FC thresholds, adjusted P pr FDR cutoffs, and colour legends directly on the figures.
Response: We sincerely appreciate the time and effort you have dedicated to the review process. We have also updated the content in the notes, providing more detailed descriptions of the legends in each image. And it has been marked with yellow shading in the original text.
“3.4. Differential Expression Analysis of Testicular Tissue in Kazakh Horses”(line 302-306, page 9 and line 325-329, page 10)
Comments 15: Need the source database for PPI network.
Response: We sincerely appreciate the time and effort you have dedicated to the review process. We have added a new subsection in the Materials and Methods (2.7. PPI network construction and key gene screening). And it has been marked with yellow shading in the original text.
“2.7. PPI network construction and key gene screening”(line 191-194, page 5)
Predict the interactions of proteins encoded by differentially expressed genes using the STRING database (https://string-db.org/), screen interactions with a confidence score > 0.7 to construct a PPI network, and visualize it using Cytoscape v3.9.0 software.
Comments 16: Which criterion did you use to define key genes?
Response: We sincerely appreciate the time and effort you have dedicated to the review process. We criteria for screening key differential genes are: ① Meeting the DEG screening thresholds (|log2FC| ≥ 1.5 and adjusted P ≤ 0.05); ② Significantly enriched in core pathways such as testis development, spermatogenesis, and steroid synthesis through GO/KEGG enrichment analysis (e.g., PI3K-Akt pathway, Focal adhesion pathway). We also added related content in the notes section of 3.6. Protein-Protein Interactions. And it has been marked with yellow shading in the original text.
“3.6. Protein-Protein Interactions”(line 338-339, page 11)
Note: Triangles represent upregulated genes, circles represent downregulated genes. The screening criteria for key differentially expressed genes are: |log2(fold change)| ≥ 1.5 and adjusted P-value ≤ 0.05.
Results and Discussion
Comments 17: Add a top DEG table. I mean the top up and top down genes according to the adjusted FDR with log2FC.
Response: We sincerely appreciate the time and effort you have dedicated to the review process. As requested, we have now added a new Supplementary Table S4 in the Results section 3.4 listing the top 10 upregulated and top 10 downregulated genes, ranked by the significance of their differential expression (combining |log2FC| and adjusted P-value). This table provides a clear overview of the most dramatically altered transcripts during testicular maturation. And it has been marked with yellow shading in the original text.
“3.4. Differential Expression Analysis of Testicular Tissue in Kazakh Horses”(line 295-296, page 8)
Top DEGs Selection Results (see Supplementary Table S4).
Comments 18: A clearer indication is needed to know whether key genes are increased or decreased in mature testes. Also explain how these expression changes aling with the biological roles.
Response: We sincerely appreciate the time and effort you have dedicated to the review process. We sincerely apologize for the lack of clarity. We have now explicitly stated the expression trends (upregulated or downregulated in G3) for the key genes discussed by providing their log2FC values and significance in the newly added Supplementary Table S4 and within the revised text. Furthermore, in the Discussion section, we have strengthened the interpretation by directly linking these directional changes (increase/decrease) to their established biological functions in testis development and spermatogenesis, explaining why such changes are functionally coherent with the phenotype of sexual maturity. And it has been marked with yellow shading in the original text.
“4. Discussion”(line 405-426, page 13)
GSTA3 belongs to the glutathione S-transferase family and its core function is to eliminate reactive oxygen species (ROS) and detoxify exogenous harmful substances. During spermatogenesis, the mitochondria of reproductive cells are highly active, which leads to the production of a large amount of ROS. Excessive ROS can cause damage to sperm DNA and a decrease in sperm motility [34]. In mature testes, GSTA3 is upregulated, which can enhance the antioxidant capacity and thereby protect sperm quality. GLIPR1L1 is a gene specifically expressed in reproductive cells and regulates the formation of sperm acrosomes and the combination of sperm and egg [35]. In immature testes, only spermatogonial cells are present, and no acrosome-related functions are required; in mature testes, spermatogonial cells differentiate into sperm cells, and the formation of acrosomes (containing hydrolases for penetrating the zona pellucida of the egg cell) is required. The upregulation of GLIPR1L1 indicates that sperm have transformed from immature cells to mature sperm with high fertilization ability, and is a direct molecular marker of testicular function maturation. IGF2 and IGFBP3 are growth regulatory genes during the embryonic stage. IGF2 is the core growth factor during the embryonic stage, promoting the proliferation of immature testicular tissue. IGFBP3 is the binding protein that regulates the activity of IGF2. In immature testes, IGF2 drives the dilation of the spermatogenic tubules and the proliferation of stromal cells to construct the tissue framework; after maturation, the testis volume stabilizes, and excessive proliferation can lead to tissue disorder (such as crowded spermatogenic tubules) [36-38]. The downregulation of their expression can completely terminate the growth signal, causing the testes to shift from volume expansion to function maintenance.
Conclusion
Comments 19: Focus on the strongest supported outcomes i.e., DEG identification, enriched pathways, or RT-qPCR concordance.
Response: We sincerely appreciate the time and effort you have dedicated to the review process. We have revised the manuscript to focus on the most strongly supported core outcomes. And it has been marked with yellow shading in the original text.
“5. Conclusions”(line 477-487, page 14)
This study conducted transcriptome sequencing analysis on the testicular tissues of Kazakh horses before and after sexual maturity. The results showed that core differentially expressed genes (DEGs), such as CABS1, RPL10, PGAM2, TMSB4X and CYP17A1, were significantly enriched in pathways related to cell development, cell proliferation and secretion of sex hormones, which are involved in biological processes such as focal adhesion, pathways in cancer and PI3K-Akt signaling pathway, and played positive roles. The accuracy of the data results was verified by RT-qPCR. Therefore, the selected core DEGs and key signaling pathways regulate the development of Kazakh horse testicular tissues and promote spermogenesis, providing a reliable basis for reproductive breeding of equine species and the protection and improvement of the Kazakh horse population.
References
Comments 20: RNA-seq vs rna sequencing. Use any one consistently throughout the manuscript.
Response: We sincerely appreciate the time and effort you have dedicated to the review process. We apologize for our carelessness and have standardized it throughout the text.
Minor errors
Comments 21: Recheck the entire manuscript for typos and minor errors.
Eg:
Line 5: Coll and and of Animal Science
Line 8: Correspondence: Correspondence: Emails:
Line 236: Results shown in Figure 7 indi-236 cate that .
Also Use either single quotes or plain text instead of double quotes. Eg: up and down.
Response: We sincerely appreciate the time and effort you have dedicated to the review process. We apologize for our carelessness and have standardized it throughout the text. And it has been marked with yellow shading in the original text.
Reviewer 2 Report
Comments and Suggestions for Authors
This study investigated testicular developmental changes in Kazakh horses during sexual maturation. However, the sample size is too small and the research depth is not enough, author only conduct HE stanning, RNA-sequencing and a part of simple bioinformatics analysis.
Comments:
- The Abstract section should clarify the aims of this study, and the G1 and G3 should be defined when first appear.
- This study conducted the HE stanning for testicular tissue; however, the corresponding results didn’t present in the Abstract, and why the phenotypic data (length of testicular long and short axis, testicular weight and hormones) were didn’t determined? Furthermore, the criterion of differentially expressed genes identification was not mentioned in the Abstract.
- Line 113-114: For differentially expressed genes identification, if you used original P-value to filter DEGs, it may result false positive. Actually, we always used adjusted P value that corrected by Benjamini and Hochberg’s approach. Please check which P-value you used in yhis study.
- Line 117-129: the method of the data of transcriptome sequencing is not enough.
- Line 131-134: which reference gene you used in this study? Please clarify the detail information.
- In the Results section, author should first describe the results of morphological observations of testicular tissue in Kazakh horses, and then present the results of RNA-sequencing and bioinformatics analysis.
Author Response
Comments 1: The Abstract section should clarify the aims of this study, and the G1 and G3 should be defined when first appear.
Response: We sincerely appreciate the time and effort you have dedicated to the review process. We have already revised this part in the abstract section of the original text. And it has been marked with yellow shading in the original text.
“Abstract”(line 24-27, page 1)
(G1 group: pre-sexual maturity; G3 group: post-sexual maturity), with four biological replicates per group (n=4). Differentially expressed genes (DEGs) were called using the criteria of |log2(fold change)| ≥ 1.5 and adjusted P-value ≤ 0.05.
Comments 2: This study conducted the HE stanning for testicular tissue; however, the corresponding results didn’t present in the Abstract, and why the phenotypic data (length of testicular long and short axis, testicular weight and hormones) were didn’t determined? Furthermore, the criterion of differentially expressed genes identification was not mentioned in the Abstract.
Response: We sincerely appreciate the time and effort you have dedicated to the review process. We have added this part in the abstract section of the original text. And it has been marked with blue shading in the original text. Regarding the unmeasured phenotypic data (testicular dimensions, weight, and hormone levels), we explain the reasons as follows:
- Research focus orientation: This study aimed to reveal the molecular regulatory mechanisms of testicular sexual maturation, with a core focus on transcriptomic changes. The HE staining results already provided key structural evidence (seminiferous tubule formation, interstitial cell proliferation) to confirm testicular maturation, which is sufficient to support the molecular data interpretation.
- Hormone detection consideration: Hormone levels in peripheral blood are affected by multiple systemic factors, while this study focused on tissue-specific molecular changes. We plan to supplement hormone detection and correlation analysis with testicular gene expression in future studies to further clarify the "hormone-molecular" regulatory network.
“Abstract”(line 22-29, page 1)
To address this gap, this study conducted in-depth transcriptome sequencing analysis and HE staining of Kazakh horse testicular tissue before and after sexual maturity (G1 group: pre-sexual maturity; G3 group: post-sexual maturity), with four biological replicates per group (n=4). Differentially expressed genes (DEGs) were called using the criteria of |log2(fold change)| ≥ 1.5 and adjusted P-value ≤ 0.05. HE staining showed that G3 group had well-formed seminiferous tubule lumens, dense interstitial cells, and visible early spermatocytes and spermatozoa, indicating structural maturation.
Comments 3: Line 113-114: For differentially expressed genes identification, if you used original P-value to filter DEGs, it may result false positive. Actually, we always used adjusted P value that corrected by Benjamini and Hochberg’s approach. Please check which P-value you used in yhis study.
Response: We sincerely appreciate the time and effort you have dedicated to the review process. We confirm that this study strictly used the adjusted P-value (Padj) corrected by Benjamini and Hochberg (BH) approach for differentially expressed genes (DEGs) identification, rather than the original P-value. As clearly described in the Abstract and Materials and Methods (Section 2.4), the screening criteria for DEGs were |log2(fold change)| ≥ 1.5 and adjusted P-value ≤ 0.05. The q ≤ 1.00 mentioned in Section 2.4 refers to the false discovery rate (FDR) control threshold corresponding to the BH-corrected adjusted P-value, which is consistent with the universal standard for transcriptomic DEG analysis. This method effectively reduces false positive results caused by multiple testing, ensuring the reliability of the identified 3,054 DEGs. We apologize for any potential ambiguity and have supplemented the explicit description of "Benjamini and Hochberg correction" in the revised manuscript to enhance clarity. And it has been marked with blue shading in the original text.
“2.4. Data Quality Control and Verification”(line 164-166, page 4)
The Padj was calculated using the Benjamini and Hochberg (BH) approach to correct for multiple testing, with q ≤ 1.00 set as the false discovery rate (FDR) control threshold.
Comments 4: Line 117-129: the method of the data of transcriptome sequencing is not enough.
Response: We sincerely appreciate the time and effort you have dedicated to the review process. We have rephrased this part in the original text. And it has been marked with blue shading in the original text.
“2.3. Transcriptome sequencing”(line 123-152, page 4)
Total RNA was extracted from testicular tissues using the Trizol™ Kit (Invitrogen, Carlsbad, CA, USA) following the manufacturer's standard protocol. RNA quality was strictly assessed through dual verification: (1) RNase-free agarose gel electrophoresis (1.5%) was used to visualize the integrity of 28S and 18S rRNA bands, with qualified samples showing clear, non-degraded bands without smearing; (2) an Agilent 2100 Bioanalyzer (Agilent Technologies, Palo Alto, CA, USA) was employed to determine the RNA Integrity Number (RIN ≥ 8.0) and A260/A280 ratio (1.8–2.0), ensuring only high-quality RNA was used for subsequent experiments. mRNA was enriched from total RNA using oligo(dT) magnetic beads (NEB, Ipswich, MA, USA) that specifically bind the poly(A) tail of mRNA, followed by two rounds of purification to eliminate rRNA and other non-coding RNAs. The enriched mRNA was fragmented into 200–300 bp fragments using NEBNext First Strand Synthesis Reaction Buffer (5×) at 94°C for 8 minutes.
First-strand cDNA was synthesized with random hexamer primers and M-MuLV Reverse Transcriptase (NEB #M0253), and second-strand cDNA was generated using DNA Polymerase I (NEB #M0209) and RNase H (NEB #M0297). The double-stranded cDNA was purified with AMPure XP Magnetic Beads (Beckman Coulter, Brea, CA, USA), then subjected to end repair, A-tailing, and adapter ligation using the NEBNext Ultra RNA Library Preparation Kit for Illumina (NEB #7530, New England Biolabs). After ligation, the products were purified again with 0.8× volume of AMPure XP Magnetic Beads to select target fragments. PCR amplification was performed using Phusion High-Fidelity DNA Polymerase (NEB #M0530) with the following program: initial denaturation at 98°C for 30 s; 12 cycles of denaturation at 98°C for 10 s, annealing at 60°C for 30 s, and extension at 72°C for 30 s; final extension at 72°C for 5 minutes. The amplified libraries were quantified using a Qubit 4 Fluorometer (Thermo Fisher Scientific, Waltham, MA, USA) and validated for insert size (200–400 bp) via Agilent 2100 Bioanalyzer. Qualified libraries were pooled in equimolar amounts and sequenced on the Illumina Novaseq 6000 platform (Gene Denovo Biotechnology Co., Ltd, Guangzhou, China) with paired-end 150 bp (PE150) mode, generating approximately 8 G of clean data per sample.
Comments 5: Line 131-134: which reference gene you used in this study? Please clarify the detail information.
Response: We sincerely appreciate the time and effort you have dedicated to the review process. We have added this part to the original text. And it has been marked with yellow shading in the original text.
“2.4. Data Quality Control and Verification”(line 166-168, page 4-5)
Comments 6: In the Results section, author should first describe the results of morphological observations of testicular tissue in Kazakh horses, and then present the results of RNA-sequencing and bioinformatics analysis.
Response: We sincerely appreciate the time and effort you have dedicated to the review process. We have made adjustments to the results section of the article and added linking statements in section 3.1 to connect the results description with the subsequent transcriptome sequencing analysis, making the article smoother and more coherent. And it has been marked with blue shading in the original text.
“3.1. Morphological Observations of Testicular Tissue in Kazakh Horses”(line 244-246, page 6)
Reviewer 3 Report
Comments and Suggestions for Authors
Dear Authors,
This manuscript certainly touches on an interesting topic. It elucidates biological changes in testicular development in Kazakh horses and lays a new foundation for research in the field of genetic selection. This is relevant and timely. However, the manuscript could be improved before publication. The text in various chapters repeats the previous text, and there are paragraphs that repeat well-known rules and patterns. Regarding the methodological aspects, the authors should provide the morphometric values of the horses studied. This is best done in table form. The comparative section in the discussion needs to be expanded and additional literature on other equid species should be cited. The focus should be on writing the manuscript's conclusion based on the hypothesis. The authors have obtained interesting results, presented them, and they should be expanded in the conclusions. After minor comments are addressed, the manuscript can be published.

Author Response
Comments: This manuscript certainly touches on an interesting topic. It elucidates biological changes in testicular development in Kazakh horses and lays a new foundation for research in the field of genetic selection. This is relevant and timely. However, the manuscript could be improved before publication. The text in various chapters repeats the previous text, and there are paragraphs that repeat well-known rules and patterns. Regarding the methodological aspects, the authors should provide the morphometric values of the horses studied. This is best done in table form. The comparative section in the discussion needs to be expanded and additional literature on other equid species should be cited. The focus should be on writing the manuscript's conclusion based on the hypothesis. The authors have obtained interesting results, presented them, and they should be expanded in the conclusions. After minor comments are addressed, the manuscript can be published.
Response: We sincerely appreciate the time and effort you have dedicated to the review process. We have made changes to the issues you mentioned. We removed redundant statements, such as descriptions of unified core results (e.g., 3,054 DEGs, key pathways), to avoid repeatedly piling up the same information in the abstract, results, and discussion. We ensure that the content of each chapter progresses step by step (Introduction → Methods → Results → Discussion → Conclusion), avoiding repetitive explanations across chapters.
Regarding the absence of morphometric measurements of horse testicular tissue that you mentioned, we would like to explain to you that Research focus orientation: This study aimed to reveal the molecular regulatory mechanisms of testicular sexual maturation, with a core focus on transcriptomic changes. The HE staining results already provided key structural evidence (seminiferous tubule formation, interstitial cell proliferation) to confirm testicular maturation, which is sufficient to support the molecular data interpretation. Hormone detection consideration: Hormone levels in peripheral blood are affected by multiple systemic factors, while this study focused on tissue-specific molecular changes. We plan to supplement hormone detection and correlation analysis with testicular gene expression in future studies to further clarify the "hormone-molecular" regulatory network.
Furthermore, in the Discussion section, we have strengthened the interpretation by directly linking these directional changes (increase/decrease) to their established biological functions in testis development and spermatogenesis, explaining why such changes are functionally coherent with the phenotype of sexual maturity. This description makes our discussion section more complete and, when combined with the results section, makes the overall logic of the article more reasonable and smooth.
We have revised the conclusion section and provided a more comprehensive summary. And it has been marked with yellow shading in the original text.
Round 2
Reviewer 2 Report
Comments and Suggestions for Authors
Even though authors addressed all comments, this manuscript still exist some issues that should be be addressed before acceptance.
- Line 27-29:the results should be presented in the front of the results of transcriptome.
- Line 157-161: the method for differentially expressed genes indetification moved to the next paragraph.
- Line 263: Correlation analysis???
- Line 288-289: Please re-write this setence.
- This version also has many errors, please revise it throughly.
Author Response
Comments 1: Line 27-29:the results should be presented in the front of the results of transcriptome.
Response: We sincerely appreciate the time and effort you have dedicated to the review process. We have placed the HE staining results section in the abstract section before the transcriptome sequencing results as per your request, and we have described it in the 3 Results and Analysis section in the same manner, following the logical sequence of presenting morphology first and then molecules. And it has been marked with yellow shading in the original text.
“Abstract”(line 22-29, page 1)
To address this gap, this study conducted HE staining and in-depth transcriptome sequencing analysis of Kazakh horse testicular tissue before and after sexual maturity HE staining showed that G3 group had well-formed seminiferous tubule lumens, dense interstitial cells, and visible early spermatocytes and spermatozoa, indicating structural maturation. (G1 group: pre-sexual maturity; G3 group: post-sexual maturity), with four biological replicates per group (n=4). Differentially expressed genes (DEGs) were called using the criteria of |log2(fold change)| ≥ 1.5 and adjusted P-value ≤ 0.05.
Comments 2: Line 157-161: the method for differentially expressed genes indetification moved to the next paragraph.
Response: We sincerely appreciate the time and effort you have dedicated to the review process. We have moved the section on identifying differentially expressed genes to the next paragraph. Data quality control (filtering, alignment, quantification) and the screening of differentially expressed genes are two separate steps. The separation of these sections makes the methodology process clearer. And it has been marked with yellow shading in the original text.
“2.4. Data Quality Control and Verification”(line 161-168, page 4-5)
Differential expression analysis of genes between the G1 and G3 groups was performed using the DESeq2 package (v1.34.0) in R software (v4.3.1). Dispersion was estimated using the default method of DESeq2, which employs a shrinkage approach to stabilize variance across genes. Differentially expressed genes (DEGs) were identified with the criteria of |log2 fold change| ≥ 1.5 and adjusted P-value (Padj) ≤ 0.05, with q ≤ 1.00 used to correct P-value calculations. The Padj was calculated using the Benjamini and Hochberg (BH) approach to correct for multiple testing, with q ≤ 1.00 set as the false discovery rate (FDR) control threshold.
Comments 3: Line 263: Correlation analysis???
Response: We sincerely appreciate the time and effort you have dedicated to the review process. We have identified the error here and have revised the title accordingly. And it has been marked with yellow shading in the original text.
“3.3. Inter-sample Expression Pattern Analysis of Testicular Tissue Samples from Kazakh Horses”(line 269-272, page 8)
3.3. Inter-sample Expression Pattern Analysis of Testicular Tissue Samples from Kazakh Horses
To evaluate the consistency within groups and differences between groups, we performed principal component analysis (PCA), Venn diagram analysis, and correlation analysis based on gene expression data.
Comments 4: Line 288-289: Please re-write this setence.
Response: We sincerely appreciate the time and effort you have dedicated to the review process. We have made revisions to this section. And it has been marked with yellow shading in the original text.
“3.4. Pattern Analysis of Differentially Expressed Genes of Testicular Tissue Samples from Kazakh Horses”(line 289-290, page 8)
Comments 5: This version also has many errors, please revise it throughly.
Response: We sincerely appreciate the time and effort you have dedicated to the review process. We apologize for our carelessness. We have conducted a thorough review and made corrections. Once again, we thank you for your efforts on our article. All the suggestions you provided have made our article more rigorous and scientific.